# Integrated Management of Chive Gnats (*Bradysia odoriphaga* Yang & Zhang) in Chives Using Entomopathogenic Nematodes and Low-Toxicity Insecticides

**DOI:** 10.3390/insects10060161

**Published:** 2019-06-05

**Authors:** Xun Yan, GuoYu Zhao, RiChou Han

**Affiliations:** 1Guangdong Key Laboratory of Animal Conservation and Resource Utilization, Guangdong Public Laboratory of Wild Animal Conservation and Utilization, Guangdong Institute of Applied Biological Resources, No. 105 Xingang Road West, Guangzhou 510260, China; yanxun@giabr.gd.cn (X.Y.); guoyu_zhao@163.com (G.Z.); 2Weifang Hongrun Agricultural Science & Technology CO., LTD, Weifang Biomedical Industry Park, Gaoxin 2nd Road, Weifang 261061, China

**Keywords:** *Bradysia odoriphaga*, *Heterorhabditis indica*, pest management, *Steinernema feltiae*, synergists

## Abstract

*Bradysia odoriphaga* is a major pest that causes damage to chive production, and which has developed resistance to highly toxic chemical insecticides. Entomopathogenic nematodes (EPN) show a high potential for *B. odoriphaga* control. This study aimed to develop an effective management method against *B. odoriphaga* larvae, using EPN with low-toxicity insecticides. Fourteen selected insecticides had no significant effects on the survival and infectivity of *Steinernema feltiae* SN and *Heterorhabditis indica* LN2. Synergistic interactions were observed for imidacloprid and osthole with *S. feltiae* SN against *B. odoriphaga* larvae. *Steinernema feltiae* SN was more effective than *H. indica* LN2 against *B. odoriphaga* at 15 and 20 °C, and the addition of imidacloprid at 1/10 recommended concentration (RC) significantly increased the efficacy of *S. feltiae* SN. The year-round occurrence of the *B. odoriphaga* larvae in chive fields treated by EPN and imidacloprid at 1/10 RC was studied. Results showed that the application of EPN with imidacloprid at 1/10 RC successfully suppressed larval populations of *B. odoriphaga* in chive fields, thus significantly increasing the yield of chive. The practical method of applying EPN–imidacloprid combinations provided a cost-effective and environmental safety strategy for controlling *B. odoriphaga* larvae in chive production, which can reduce the usage of toxic chemical insecticides.

## 1. Introduction

The chive gnat, *Bradysia odoriphaga* Yang and Zhang (Diptera: Sciaridae), is a major insect pest in chive production in China [1]. *Bradysia odoriphaga* feeds on plants from seven families and on more than thirty species, and it also causes production losses in mushroom sheds [2]. Larvae of *B. odoriphaga* aggregate in the soil and feed on the roots and stems of the plant, causing plants to stunt or even die [3]. *Bradysia odoriphaga* has 4–6 overlapping generations per year, and the peak damage occurs in spring and autumn [2,4], which leads to chive production losses of 30% to 80% in North China [5].

Larvae of *B. odoriphaga* are cryptic in the soil, and therefore much more toxic insecticides are used by the farmer to maintain yields, which has led to heavy insecticide residues in the chives and the soil [6]. Over-use of toxic insecticides has also led to the development of insecticide resistance in *B. odoriphaga*, and it has developed widespread resistance to organophosphorus insecticides such as chlorpyrifos and phoxim [7,8]. New effective and environmentally friendly methods for controlling *B. odoriphaga* are urgently needed.

Entomopathogenic nematodes (EPN) of the genera *Steinernema* and *Heterorhabditis* (Rhabditida: Steinernematidae and Heterorhabditidae) have been successfully used to control some soil-dwelling and boring insect pests, because they can actively search for their hosts [9,10,11,12]. Earlier studies have showed that EPN had the potential to control *B. odoriphaga*. Two EPN species, *S. feltiae* and *S. hebeiense*, were found to have the ability to suppress the larval population of *B. odoriphaga* in chive fields to a level that is similar to phoxim treatment [3]. *Heterorhabditis indica* could also control the *B. odoriphaga* larvae in chive field, and the control effect could last for at least three months after a one-time application, which was more effective than phoxim application [13]. Although EPN have the potential to control *B. odoriphaga*, their efficacy in the field might not be stable, due to the influences from different environmental factors [12,14]. Further, when compared to common chemical insecticides, the cost of EPN remains too high for most of the chive growers in China [15]. Combining EPN with insecticides has contributed to the control of a number of economically important crop pests [15,16,17,18,19]. Therefore, the improvement of EPN efficacy by combining EPN with low-toxicity insecticides is an ideal strategy for the effective control of *B. odoriphaga* larvae.

Previous studies have indicated that EPN are compatible with different kinds of chemical insecticides, and some insecticides have been reported to be synergistic with EPN against white grub (*Holotrichia oblita*, *Popillia japonica*, *Exomala* (*Anomala*) *orientalis*, *Cyclocephala* spp., etc), whitefly, (*Bemisia tabaci*) and leafminer (*Liriomyza huidobrensis*, *Tuta absoluta*) [15,16,17,18,19,20,21,22,23,24,25,26,27]. The effect of *S. feltiae* SN with thiamethoxam to control *B. odoriphaga* was studied in chive fields in Tai’an, China [28]. However, better and cheaper synergistic insecticides should be screened before a practical method that involves the combination of EPN in insecticides to control *B. odoriphaga* in chive field is established. Furthermore, the dynamics of *B. odoriphaga* larvae in chive fields after EPN–insecticide applications should also be investigated for a longer period that covers the entire season of chive production, and the chive yield and production cost also needs to be compared with the control and insecticide treatments. The purpose of this study is to screen compatible insecticides that showed synergistic effects with EPN against larvae of *B. odoriphaga*, and to develop an environmentally friendly and cost-effective method of controlling *B. odoriphaga* in chive production.

## 2. Materials and Methods

### 2.1. Nematodes

Infective juveniles (IJs) of *S. feltiae* SN and *H. indica* LN2 were provided by Weifang Hongrun Agriculture Science and Technology Co., LTD, China. Survival of IJ were examined under a stereoscopic microscope before use, and they were considered dead when they showed no response after probing with a needle [26]. The background mortality of the IJ was below 10% throughout the study. All laboratory bioassays were repeated twice, using different batches of nematodes.

### 2.2. Insects

Larvae of *B. odoriphaga* were collected from a chive field located at Baijia Village, Gaozhai County, Zhangqiu District in Jinan city. The chive gnats were reared in 9 cm diameter Petri dishes, furnished with two layers of 9 cm diameter filter paper and a 2 cm length of chive stem, in the laboratory for 3–4 days. The fourth instar larvae of *B. odoriphaga* were used for bioassays [3,29].

The ninth to 11th instar larvae of yellow mealworm, *Tenebrio molitor* L., were purchased from the Binzhou Xinchong yellow mealworm breeding factory, China. The yellow mealworms were reared at 25 ± 1 °C for 2 days after purchase, and used for the bioassay [30].

### 2.3. Laboratory Bioassay

#### 2.3.1. Insecticides

Three kinds of insecticide used for the bioassay are listed in Table 1, which included chemical insecticides, botanical insecticides, and insect growth regulators. All of the insecticides were commercial formulations purchased from the Chinese market, and they are commonly used in Chinese vegetable production.

#### 2.3.2. Effect of Insecticides on EPN Survival

Insecticides were diluted and added to each 9 cm Petri dish, with each dish containing 1000 IJs in 10 mL to a final concentration corresponding to the recommended field concentration (RC) and 1/10 RC. Infective juveniles in water were used as controls. Each treatment had three replicates (dishes). The Petri dishes were incubated at 25 ± 1 °C in the dark for 24 h. Three 100 μL samples were then drawn from each dish and mortality of the IJs was calculated. Infective juveniles were considered dead when they showed no response after probing with a needle [26].

#### 2.3.3. Effect of Insecticides on EPN Infectivity

The nematode suspension in 1.5 mL containing 500 live IJs were added into a 9 cm Petri dish lined with two layers of 9 cm filter paper (Xinhua, China). Ten ninth to 11th *T. molitor* larvae (body lengths 2–3 cm) were then placed in each dish [26]. Treatments contained EPN alone, EPN–insecticide combinations and insecticide dilutions at the respective concentrations (as described above). Water treatment was used as control. Each treatment had three replicates (dishes). The dishes were sealed with Parafilm and incubated at 25 ± 1 °C, 70% RH for 72 h. Mortality of the *T. molitor* larvae was checked. Dead larvae were incubated in clean Petri dishes and dissected three days later to estimate the rate of infection by EPN [9].

#### 2.3.4. Virulence of EPN–Insecticide Combinations against *B. odoriphaga*

One 2 cm length stem of chive was put into each 9 cm Petri dish furnished with two layers of 9 cm filter paper. Nematodes were treated with the selected insecticides and adjusted to 250 IJ·mL^−1^. Suspensions of 1 mL of insecticide at 1/10 RC, EPN–insecticide (1/10 RC) combination, or EPN were added to each dish. Ten *B. odoriphaga* larvae at fourth instar were gently placed on the filter paper by a brush. Water treatment was used as the control. Each treatment had three replicates (dishes). The dishes were sealed with Parafilm and incubated at 25 ± 1 °C for 72 h. The mortality of the *B. odoriphaga* larvae was checked every 24 h. Dead larvae were incubated in clean Petri dishes at 25 ± 1 °C, 70% relative humidity (RH), and dissected two days later, to estimate the rate of infection by EPN [3].

#### 2.3.5. Effect of Temperature on the Virulence of EPN–Imidacloprid Combinations against *B. odoriphaga*

Insecticides that showed potential synergistic effects on EPN were used to further test the effects of temperature on the virulence of the EPN-insecticide combinations against the *B. odoriphaga* larvae. The method was the same as described above [3]; besides that, the dishes were incubated at 15 ± 1, 20 ± 1 and 25 ± 1 °C for 72 h.

### 2.4. Field Experiment

Two experiments were conducted in chive fields that was naturally infested with *B. odoriphaga* in 2016–2017, in Zhangqiu, Jinan City and in Weifang City in Shandong Province, China. Soil samples were baited with *T. molitor*, using the method described previously by Liu et al. [31], and no native EPN populations were recovered from the experimental field. Larval populations of the chive gnat were estimated one day before the field experiment. Five soil samples (0.08 m^2^, depth = 20 cm) containing the chive plants were individually taken from the experimental field, and the number of the larval and pupal stages of *B. odoriphaga* in the soil and on the plants was counted. The population of the *B. odoriphaga* larvae was expressed as number of larvae·m^−2^. In both experiments, the plant density was 36 plants·m^−2^. Imidacloprid were applied at 4.5 g·ha^−1^, together with *H. indica* LN2 at 3.0 × 10^9^ IJ·ha^−1^ (LN2 + IMI), *S. feltiae* SN at 3.0 × 10^9^ IJ·ha^−1^ (SN + IMI), or *H. indica* LN2 and *S. feltiae* SN both at 1.5 × 10^9^ IJ·ha^−1^ (LN2 + SN + IMI). Water without nematodes or insecticides was used as the negative control. Each treatment had five replicates (plots) arranged in a randomized complete block design. The application of nematodes was done on cloudy days or after 4:00 pm, to avoid strong UV radiation.

The first experiment was conducted at Baijia Village in Zhangqiu on 8 October, 2016. Experimental plot was 148 m^2^ (74 m × 2 m), with a 0.6 m buffer space between plots. The density of the chive gnats in the field was 277.50 ± 29.42 larvae·m^−2^. Chlorbenzuron was applied at RC as the positive control. Nematode and insecticide suspensions were applied by using the irrigation system (with nozzle size of 6.5 mm in diameter). All treatments were applied three times on the date of the experiment, 27 March and 23 September 2017. Chlorbenzuron was applied one extra time, on 17 December, 2016. Chive was harvested on 17 December 2016 and 3 January, 2017. Imidacloprid were also applied at RC, but it showed no difference to the control on the suppression of the chive gnat in the first 1.5 months, so no further survey was conducted on the plots treated with imidacloprid.

The second experiment was conducted at Daotian Town in Weifang on 24 October, 2016. The experimental plot was 74 m^2^ (37 m × 2 m), with a 0.6 m buffer space between the plots. The density of chive gnats in the field was 283.75 ± 36.58 larvae·m^−2^. Phoxim (Shengbang Lunan Pesticide Co., Ltd. of Shandong, recommended field rate 1125 mL·ha^−1^) was applied at RC as the positive control. Nematode and insecticide suspensions were applied by using a sprayer with a flat-fan nozzle at 206 kPa (with nozzle size of 6.5 mm in diameter), without a screen. All treatments were applied twice on the date of the experiment, and on 30 September, 2017. Phoxim was applied twice more, on 23 December, 2016, and 9 February, 2017. The chives were harvested on 23 December, 2016, and 23 January and 28 February, 2017.

A field survey was conducted every 10 or 20 days until 360 days after the first application, based on the dynamics of *B. odoriphaga* in the field. Five chive plants were sampled from each plot as described above, to estimate the number of larvae and pupae from each sample. All of the fresh chive plants harvested from each plot were weighed, and the yields of chive were calculated as kg·m^−2^.

### 2.5. Cost Estimation of EPN–Imidacloprid Applications

The costs for *B. odoriphaga* control in chive production were estimated based on the retail prices of the EPN and the insecticides in October 2017. The average chive yield was calculated as kg·ha^−1^. The costs for different treatments were estimated. The net profit was estimated based on the average price of chives at three harvest times, and the cost per ha from the different treatments [15].

### 2.6. Statistical Analysis

Mortality rates of *B. odoriphaga* and *T. molitor* were corrected by control mortality using Abbott’s formula [32]. Percentage data were arcsine square root-transformed prior to statistical analysis performed with SPSS 16.0 software (SPSS Inc., Chicago, IL, USA). Means were separated using Tukey’s test. Differences among the means were considered significant at *p* < 0.05. Synergistic, additive, or antagonistic interactions between the agents in the combination treatments were determined by using a χ^2^ test with the control-corrected data [15].

## 3. Results

### 3.1. Effect of Insecticides on EPN Survival

The mortality rates of *S. feltiae* SN and *H. indica* LN2 exposed to two concentrations of the tested insecticides are shown in Figure 1. No significant differences in the mortality rate were found among the different insecticide treatments and the control, for both EPN isolates (F ≤ 1.171, df = 14, 30, *p* ≥ 0.345). The mortality rates of the two EPN isolates after treatment with 14 insecticides at RC and 1/10 RC were < 2%. All the insecticides were used to test their effects on EPN infectivity.

### 3.2. Effect of Insecticides on EPN Infectivity

The corrected mortality rates of the *T. molitor* larvae exposed to *S. feltiae* SN and *H. indica* LN2 treated by two concentrations of different insecticides are shown in Figure 2. The corrected mortality rates for the larvae caused by EPN and treated with different insecticides differed significantly, but no significant difference was found between the corrected mortality rates of the larvae as caused by insecticide-treated and untreated nematodes, for each insecticide (F ≥ 2.412, df = 14, 30, *p* ≤ 0.021). The corrected mortality rates of the larvae treated by *S. feltiae* SN ranged from 53.33 ± 3.33% to 96.67 ± 3.33%, and from 40.00 ± 5.77% to 93.33 ± 6.67% by *H. indica* LN2. Both *S. feltiae* SN and *H. indica* LN2 treated with imidacloprid caused the highest mortality of the *T. molitor* larvae when compared with the other insecticides at the same concentration.

### 3.3. Virulence of EPN–Insecticide Combinations against *B. odoriphaga*

The corrected mortality rates of the *B. odoriphaga* larvae treated with EPN–insecticide combinations are shown in Table 2. Combinations of different insecticides with *S. feltiae* SN did not cause significant differences in the mortality rate of the *B. odoriphaga* larvae when compared with *S. feltiae* SN, after treating the larvae for 48 h and 72 h (F = 3.090 and 1.331, df = 14, 30, *p* = 0.005 and 0.270, respectively). However, at 24 h, the combination of imidacloprid or osthole with *S. feltiae* SN caused a 93.33 ± 3.33% mortality rate in the *B. odoriphaga* larvae, which was significantly higher than that caused by *S. feltiae* SN (F = 6.376, df = 14, 30, *p* < 0.001). The χ^2^ test showed that synergistic interactions were observed for the imidacloprid–*S. feltiae* SN (χ^2^ = 4.03) and osthole–*S. feltiae* SN (χ^2^ = 4.57) combinations. Different insecticide–*H. indica* LN2 combinations did not cause significantly mortality of the *B. odoriphaga* larvae, when compared with *H. indica* LN2 at three tested times (F ≤ 3.877, df = 14, 30, *p* ≥ 0.001). No synergistic interactions were observed for different insecticides with *H. indica* LN2 (χ^2^ ≤ 1.822).

### 3.4. Effect of Temperature on the Virulence of EPN–Imidacloprid Combinations against *B. odoriphaga*

Corrected mortality rates of the *B. odoriphaga* larvae treated with imidacloprid-EPN combinations at different temperatures are shown in Figure 3. At a low temperature of 15 °C, treatments of *H. indica* LN2 and *H. indica* LN2–imidacloprid did not cause the death of the *B. odoriphaga* larvae. Treatments containing *S. feltiae* SN caused significantly higher corrected mortality rates of the *B. odoriphaga* larvae (F ≥ 88.893, df = 4, 10, *p* < 0.001), which were > 93.33% after 72 h. Treatments of *H. indica* LN2 and *H. indica* LN2–imidacloprid showed poor virulence against the *B. odoriphaga* larvae at 20 °C, with the corrected mortality rates of the *B. odoriphaga* larvae being significantly lower than those in treatments containing *S. feltiae* SN at three checking times (F ≥ 16.631, df = 4, 10, *p* < 0.001). The corrected mortality rates of the *B. odoriphaga* larvae caused by treatments containing *S. feltiae* SN were > 90.00% after 48 h. The corrected mortality rates of the *B. odoriphaga* larvae caused by *H. indica* LN2 and *H. indica* LN2–imidacloprid increased at 25 °C, but they was still significantly lower after 24 h and 48 h (F ≥ 5.027, df = 4, 10, *p* ≤ 0.018). The corrected mortality rates of the *B. odoriphaga* larvae were all > 90.00% after 72 h at 25 °C.

### 3.5. Effects of EPN–Imidacloprid Applications on a *B. odoriphaga* Larval Population in the Field

Larval populations of *B. odoriphaga* in the field during the experimental period are shown in Figure 4. In the experiment conducted in Zhangqiu, larval populations in the plots with treatments containing *S. feltiae* SN or chlorbenzuron decreased over time during the first 60 days, and they were significantly lower than those in the plots treated with *H. indica* LN2-imidacloprid or water (F ≥ 6.405, df = 4, 20, *p* < 0.001). Larval population in the plots treated with chlorbenzuron increased significantly at 70 days when compared with the treatments containing *S. feltiae* SN (F = 17.876, df = 4, 20, *p* < 0.001). One more application of chlorbenzuron at RC was performed to protect the chives. The larval population decreased again until 110 days, but it then increased again in the plots treated with chlorbenzuron to the same level as in the water control. Larval populations in the plots with treatments containing *S. feltiae* SN remained significantly lower than those in the other plots (F = 23.711, df = 4, 20, *p* < 0.001). Protective films used to cover the plants were removed at 120 days, and the larvae had hardly recovered from the soil because of the cold weather. Larval populations increased later on, with the increasing temperature, so a re-application of all treatments were conducted on 27 March, 2017. After the second application of all treatments, larval populations in the plots with treatments containing *S. feltiae* SN or chlorbenzuron were significantly lower than those with other treatments (F ≥ 12.863, df = 4, 20, *p* < 0.001). By late May, the high temperature had resulted in a lower larval population in the field. The larval population increased again when the temperature decreased in October. The reapplication of all treatments was required to protect the chive production for next year at this time. The highest larval population recorded in the field was 52.97 larvae·m^−2^ after treatment, but this was 27.39 larvae·m^−2^ in the plots with treatments containing *S. feltiae* SN.

In the experiment conducted in Weifang, the dynamics of the larval population of *B. odoriphaga* in the field were similar to those in the experiment conducted in Zhangqiu. However, no obvious decrease of the larvae in the field was observed in February, 2017. Chives were planted under the protective film in a green house (so-called double-sheds). The temperature could be maintained in the green house after the removal of the inner protective film after the third harvest of chives, which led to a continously high larval population in the control. EPN–imidacloprid combinations sucessfully suppressed the larval population in this field. Larval populations were significantly lower in the plots with EPN–imidacloprid treatments than those with phoxim or water at most sample times (F ≥ 21.859, df = 4, 20, *p* < 0.001). No significant differences in larval populations was found among plots with phoxim and EPN–imidacloprid treatments at sample times shortly after phoxim application, and the larval populations in the plots with these treatments were significantly higher than those with water (F ≥ 3.146, df = 4, 20, *p* < 0.05). No significant differences in larval populations were found in the plots with different treatments from June to October 2017 (F ≤ 1.097, df = 4, 20, *p* ≥ 0.385). The highest larval population recorded in the control plots was 78.71 larvae·m^−2^, and 24.88 larvae·m^−2^ in plots with EPN–imidaclprid treatments.

### 3.6. Effects of EPN–Imidacloprid Applications on Chive Yields

Chive weights from different treatments in the two experiments are shown in Figure 5. In the experiment conducted in Zhangqiu, chive yields in the plots with treatments containing *S. feltiae* or chlorbenzuron were > 3.55 kg·m^−2^, which were significantly higher than that in the plots treated with *H. indica* LN2–imidacloprid (2.88 kg·m^−2^) or water (2.68 kg·m^−2^) (F = 25.383, df = 4, 20, *p* < 0.001). In the experiment conducted in Weifang, average yields of chive in the plots with EPN–imidacloprid treatments or phoxim were 4.25 to 4.56 kg·m^−2^, which were significantly higher than that with water (2.44 kg·m^−2^) (F = 89.006, df = 4, 20, *p* < 0.001). The average yields of chive harvested from the treated plots in the experiment conducted in Weifang were higher than that in Zhangqiu, although the yields in the control plots were higher in Zhangqiu than in Weifang.

### 3.7. Cost Estimation of the Integrated Management Strategy

The cost, yield of chive, and the income of the farmers by using EPN–imidacloprid and chemicals in one hectare for *B. odoriphaga* control in chive production were calculated and shown in Table 3. In the experiment conducted in Zhangqiu, the profit was the highest in the field treated with *S. feltiae* SN-imidacloprid and *H. indica* LN2-*S. feltiae* SN-imidacloprid, because the chive price was higher from EPN-treated fields than chlorbenzuron-treated field. In the experiment conducted in Weifang, the profit was similar for different treatments except for the water control, because the chive price was the same from field with different treatments.

## 4. Discussion

Evaluating the compatibility of EPN with different insecticides, and screening additive or synergistic insecticides for EPN provide the basis for the integrated management of target pests by using EPN with insecticides in the field. The co-application of EPN with insecticides can increase the control efficacy against a target pest, and reduce the cost of EPN applications and the dependence on chemical insecticides, thus contributing to slowing down the development of insecticide resistance and preventing adverse effects on human beings and the environment [9,15]. Fourteen insecticides tested in the present study were all compatible with *S. feltiae* SN and *H. indica* LN2, and had no adverse effect on the infectivity of the two EPN isolates against *T. molitor* larvae. The results suggested that the tested insecticides could be co-applied with *S. feltiae* SN and *H. indica* LN2. *Steinernema feltiae* was compatible with the azadirachtin product tested in this study, and it was reported to be compatible with some but not all formulations of azadirachtin [18,22]. The difference among different studies may be related to different formulations of the tested compounds and the experimental design [26]. As very few studies were reported on the compatibility of *H. indica* LN2 with insecticides, our results provided a reference for the combined use of *H. indica* LN2 with different insecticides.

In the experiment testing the effect of insecticides on EPN infectivity, imidacloprid was found to enhance the infectivity of *S. feltiae* SN and *H. indica* LN2 against *T. molitor* larvae. Results of the bioassay on *B. odoriphaga* further confirmed the synergistic effect of imidacloprid with *S. feltiae* SN. Imidacloprid belongs to a neonicotinoid, and it is a systemic insecticide that acts as an insect neurotoxin. Imidacloprid was widely used in pest control because of its high efficacy, relatively low vertebrae toxicity, low application rates, and long systemic persistence [21]. Imidacloprid was reported to interact synergistically with *H. bacteriophora*, *H. megidis*, *H. marclatus*, *S. glaseri*, and *S. feltiae* against different species of white grub [21,33,34] and sweet potato whitefly [17]. Synergistic interactions were not found between *H. indica* LN2 and imidacloprid against *B. odoriphaga* in this study. Koppenhöfer et al. also found only additive effects of imidacloprid with *S. kushidai* against white grubs [35]. The major factor that is responsible for synergistic interactions between imidacloprid and EPN appears to be the general disruption of normal nerve function due to imidacloprid, which resulting in drastically reduced activity by the grubs. This sluggishness facilitates host attachment of the nematodes. Grooming and evasive behavior in response to nematode attack was also reduced in imidacloprid-treated grubs [35]. Osthole is a botanical insecticide that showed toxicity to the third instar larvae of *B. odoriphaga* [36]. Although osthole also showed synergistic interactions with *S. feltiae* SN against *B. odoriphaga*, it was not used for the field experiment, considering the cost of its application. The cost of a one-time application of osthole is three times as much as that of imidacloprid. Using imidacloprid as a synergist to EPN helps with a cost-effective method to control the chive gnat to be developed.

The mortality rates of *B. odoriphaga* larvae treated by the two EPN isolates in the study were ≥ 90%, which suggested that the EPN products from solid cultures were as active as the nematodes from in vivo cultures used by Ma et al. [3]. However, for the same EPN isolates from another producer, the mortality rates of *B. odoriphaga* larvae were ≤ 50%, even when the IJ were applied at high concentrations of 200 IJ·larvae^-1^ [28]. The difference in pathogenicity of the same EPN isolate from different production companies probably is due to the IJ quality [37]. The stability of the nematodes would be impacted during shipping and storage [38]. It is important to ensure that the EPN products are still at a high quality when the end users receive and apply them. This is the guarantee for the effective control of pests by EPN and other live biological control agents.

Temperature plays an important role in the motility, infectivity, development, and survival of EPN [39]. Infective juveniles of EPN slow down their movement and metabolism at low temperatures, which will affect their infectivity against the target host [40]. *Bradysia odoriphaga* can survival at low temperatures, and it can be active at 10 °C. As the optimal temperature for *B. odoriphaga* is 15–24 °C [41], it is important to know whether the selected EPN is still active against the *B. odoriphaga* larvae when the soil temperature decreases. The results showed that *H. indica* LN2 and *H. indica* LN2–imidacloprid combinations were not as effective as treatments containing *S. feltiae* SN at 15 and 20 °C, suggesting that *H. indica* LN2 was not as active as *S. feltiae* SN at 15 and 20 °C. The addition of imidacloprid did not increase the infectivity of *H. indica* LN2 against *B. odoriphaga* at these two temperatures. When temperature increased to 25 °C, the treatments of *H. indica* LN2 and *H. indica* LN2–imidacloprid combination increased the efficacy against *B. odoriphaga*, suggesting that *H. indica* LN2 was more active at a high temperature. *Heterorhabditis indica* LN2 has also been reported to show poor infectivity against *Phyllotreta striolata* and *Agrotis ipsilon* larvae at low temperature [11,42]. *Heterorhabditis indica* LN2 was first isolated in India, and it has adapted itself to a relatively high temperature where it was recovered [43], which might explain the poor performance of *H. indica* LN2 against pests at low temperatures.

It is the first time to report a year-around occurrence of the *B. odoriphaga* larvae after treatment with EPN in chive fields. Results showed that treatment of *S. feltiae* SN–imidacloprid and *S. feltiae* SN-*H. indica* LN2–imidacloprid combinations constantly suppress the larval population of *B. odoriphaga*, and this suppression effect could last for almost 11 months in both field experiments. In an experiment conducted in Weifang, treatment of *H. indica* LN2–imidacloprid combination showed similar suppression effects on the larval population of *B. odoriphaga*. While in the experiment conducted in Zhangqiu, treatment of *H. indica* LN2–imidacloprid combination did not show significant suppression effects on the larval population of *B. odoriphaga* when compared with treatments containing *S. feltiae* SN. Chives were planted under different modes in the two fields. Chives in the field in Weifang were planted under double-sheds. The temperature in the soil was relatively constant and high, so *H. indica* LN2 could perform well. While in the field in Zhangqiu, the chives were planted under only one layer of protective film. The temperature in this field was not as high as that in Weifang, and the films were removed from February to late October. This might explain why *H. indica* LN2 could not perform as well in the field in Weifang.

Cost and profit are the main factors that farmers or companies concerned about, when EPN are applied. In Zhangqiu, chives harvested from EPN–imidacloprid treatments were sold at a higher price than those from chlorbenzuron treatment. Although the cost of the EPN–imidacloprid treatments was 2.0 to 3.6 times higher than that of the chlorbenzuron treatment, and the yield of the chive did not vary much, the profits from the EPN–imidacloprid treatments were 1.7 to 2.4 times as much as those from the chlorbenzuron treatment. The ecological and economic effects were superior in chives produced by EPN–imidacloprid treatments than by chlorbenzuron treatment. In Weifang, where the price of chives was the same in different treatments, the profit was still slightly higher with the EPN–imidacloprid treatments than with the phoxim treatment. The application of EPN–imidacloprid combinations could provide a competitive method for chemical control against *B. odoriphaga* in chive production, regardless of the ecological effects. If chives protected by EPN–imidacloprid treatments could be sold at a higher price than by insecticides, the profit will significantly increase.

## 5. Conclusions

The present study developed a cost-effective management method to control *B. odoriphaga* in chive production. In chive planting fields furnished with one layer of protective film, two rounds of application of *S. feltiae* SN–imidacloprid or *S. feltiae* SN + *H. indica* LN2 + imidacloprid in one year is recommended for effective *B. odoriphaga* control in the field. In chive planting in a greenhouse furnished with one layer of protective film, two applications of *S. feltiae* SN or *H. indica* LN2 with imidacloprid were both recommended for effective *B. odoriphaga* control.

## Figures and Tables

**Figure 1 insects-10-00161-f001:**
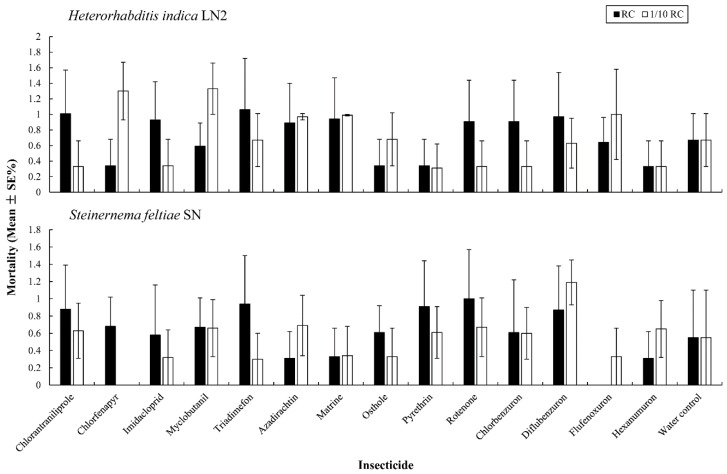
Mortality rates of *Steinernema feltiae* SN and *Heterorhabditis indica* LN2, 72 h after exposure to different insecticides at the recommended field concentration (RC) and 1/10 RC. The bars do not differ significantly at the same insecticide concentration (*p* > 0.05, Tukey’s test).

**Figure 2 insects-10-00161-f002:**
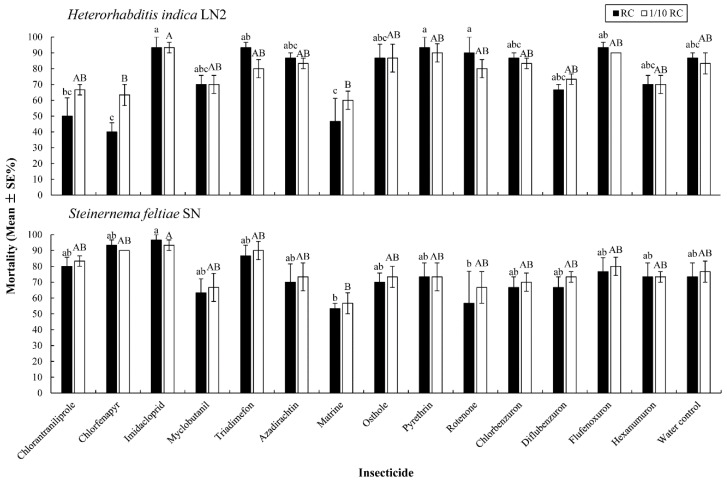
The corrected mortality rates of the *Tenebrio molitor* larvae, 72 h after exposure to *Steinernema feltiae* SN and *Heterorhabditis indica* LN2 in different insecticides, at the recommended field concentration (RC) and 1/10 RC. Bars with different uppercase or lowercase letters indicate significant differences among the different insecticides for RC or 1/10 RC, respectively (*p* < 0.05, Tukey’s test).

**Figure 3 insects-10-00161-f003:**
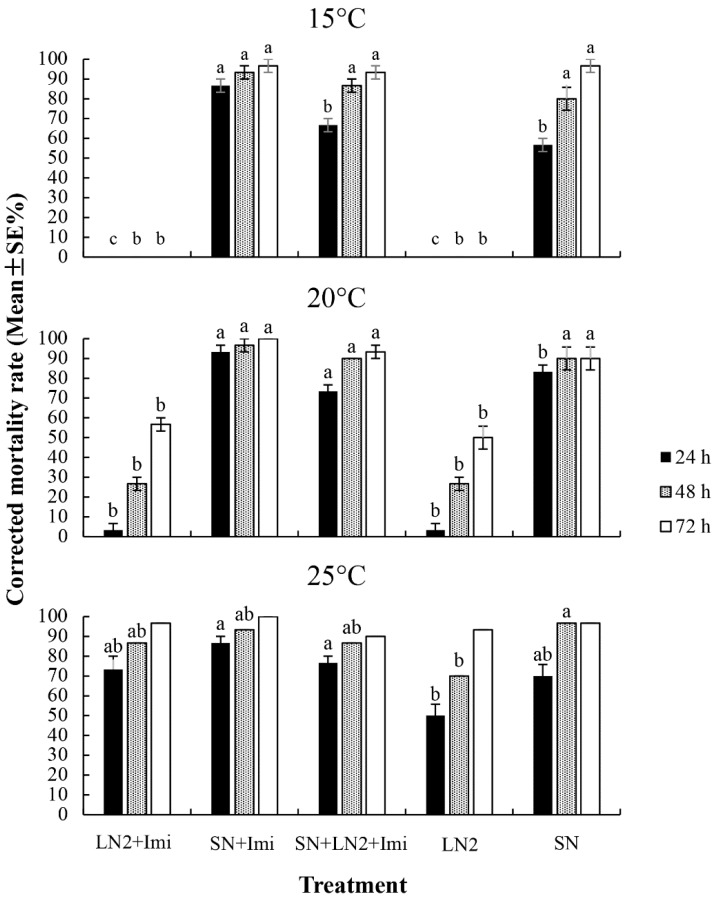
The corrected mortality rates of the *Bradysia odoriphaga* larvae treated with entomopathogenic nematodes (EPN) and EPN–imidacloprid combinations for 24, 48, and 72 h. LN2: *Heterorhabditis indica* LN2 applied at 3.0 × 10^9^ IJ·ha^−1^; SN: *Steinernema feltiae* SN applied at 3.0 × 10^9^ IJ·ha^−1^; Imi: imidacloprid at 1/10 recommended field rate (4.5 g·ha^−1^); SN + LN2: both *H. indica* LN2 and *S. feltiae* SN applied at 1.5 × 10^9^ IJ·ha^−1^. Bars with the same letter do not differ significantly at the same checking time (*p* < 0.05, Tukey’s test).

**Figure 4 insects-10-00161-f004:**
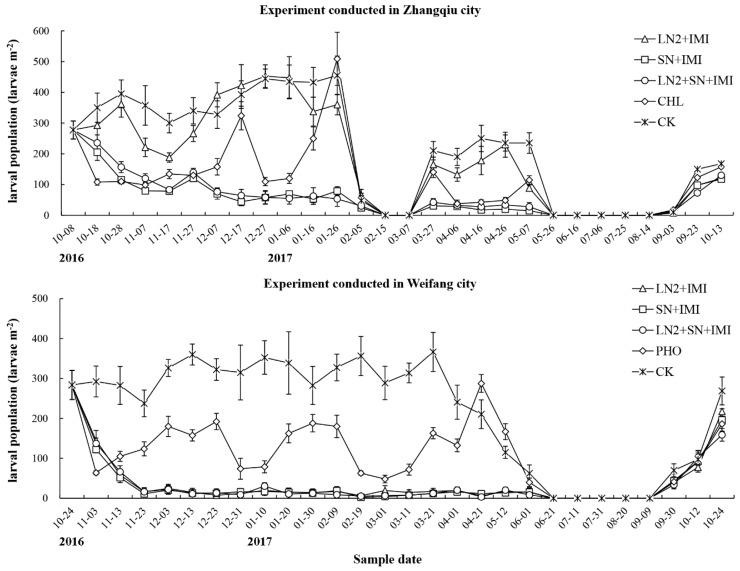
The larval population of *Bradysia odoriphaga* recovered from field surveys in the experiments conducted in Zhangqiu and Weifang city in China, in 2016–2017. LN2: *Heterorhabditis indica* LN2 applied at 3.0 × 10^9^ IJ·ha^−1^; SN: *Steinernema feltiae* SN applied at 3.0 × 10^9^ IJ·ha^−1^; Imi: imidacloprid at 1/10 recommended field rate (4.5 g·ha^−1^); SN + LN2: both *H. indica* LN2 and *S. feltiae* SN applied at 1.5 × 10^9^ IJ·ha^−1^; CHL: chlorbenzuron applied at the recommended field rate of 15 mg·kg^−1^; PHO: phoxim applied at the recommended field rate of 1125 mL·ha; CK, water control.

**Figure 5 insects-10-00161-f005:**
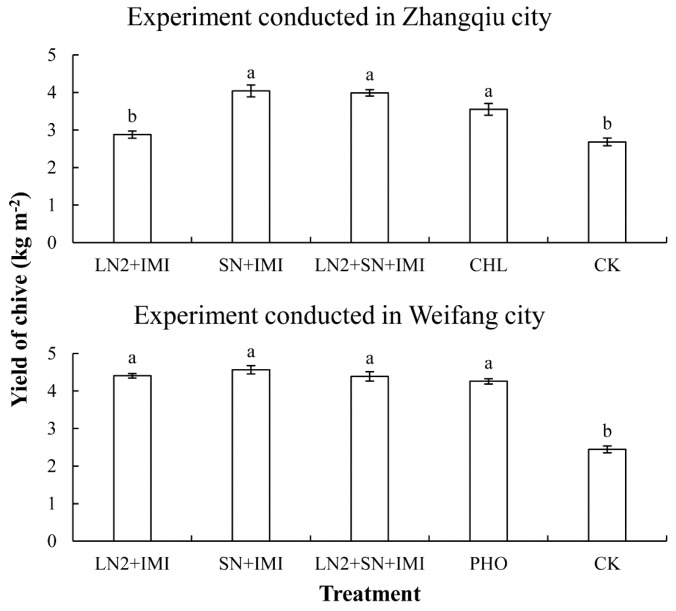
Average yields of chive in the plots with different treatments in the experiments conducted in Zhangqiu and Weifang city in China, in 2016–2017. LN2: *Heterorhabditis indica* LN2 applied at 3.0 × 10^9^ IJ·ha^−1^; SN: *Steinernema feltiae* SN applied at 3.0 × 10^9^ IJ·ha^−1^; Imi: imidacloprid at 1/10 recommended field rate (4.5 g·ha^−1^); SN + LN2: both *H. indica* LN2 and *S. feltiae* SN applied at 1.5 × 10^9^ IJ·ha^−1^; CHL: chlorbenzuron applied at the recommended field rate of 15 mg·kg^−1^; PHO: phoxim applied at the recommended field rate of 1125 mL·ha^−1^; CK, water control. Bars with the same letter do not differ significantly (*p* < 0.05, Tukey’s test).

**Table 1 insects-10-00161-t001:** Low-toxicity insecticides used in the study.

Insecticide ^1^	Main Component	Manufacturer	Recommended Field Rate
Chemical insecticide			
Chlorantraniliprole SC 20%	chlorantraniliprole	DuPont USA	150 mL·ha^−1^
Chlorfenapyr SC 10%	chlorfenapyr	BASF	750 mL·ha^−1^
Imidacloprid WG 70%	imidacloprid	Bayer	45 g·ha^−1^
Myclobutanil EC 12.5%	myclobutanil	Kaifeng Tianwei Biochemical Co., Ltd.	450 mL·ha^−1^
Triadimefon GR 15%	triadimefon	Sichuan Guoguang Agriculture and Chemistry Co., Ltd.	900 g·ha^−1^
Botanical insecticide			
Azadirachtin SC 0.3%	azadirachtin	Chengdu Green the High-Tech Co., Ltd.	2250 mL·ha^−1^
Matrine SC 0.5%	matrine	Beijing Green Agricultural Science and Technology Group Co., Ltd.	1350 mL·ha^−1^
Osthole EC 1.0%	cnidium lactone	Suke Agro-chemical of Jiangsu Province Co., Ltd.	3000 g·ha^−1^
Pyrethrin SC 1.5%	pyrethrin	Beijing Kingbo Biotech Co., Ltd.	2400 mL·ha^−1^
Rotenone SC 7.5%	rotenone	Beijing Kingbo Biotech Co., Ltd.	975 mL·ha^−1^
Insect growth regulator			
Chlorbenzuron SC 25%	chlorbenzuron	HONOR-BIO	150 mg·kg^−1^
Diflubenzuron SC 20%	diflubenzuron	Anlin Biochemical Co., Ltd.	133 mg·L^−1^
Flufenoxuron SC 2.5%	flufenoxuron	BASF	750 mL·ha^−1^
Hexanumuron SC 5%	hexanumuron	Jiukang Biological Science and Technology Development Co., Ltd.	120 g·ha^−1^

^1^ EC, emulsifiable concentrate; GR, granules; SC, suspension concentrate; WG, water-dispersible granules.

**Table 2 insects-10-00161-t002:** The corrected mortality rate (± SE) of *Bradysia odoriphaga* larvae treated with different entomopathogenic nematode and insecticide combinations after 24, 48 and 72 h exposure to *Steinernema feltiae* SN and *Heterorhabditis indica* LN2 in different insecticides at the diluted recommended field concentration (1/10 RC).

Insecticide	Corrected Mortality Rate (Mean ± SE %) ^1^					
	*Steinernema feltiae* SN			*Heterorhabditis indica* LN2		
	24 h	48 h	72 h	24 h	48 h	72 h
Chlorantraniliprole	66.67 ± 3.33 bc	73.33 ± 3.33 b	96.67 ± 3.33	46.67 ± 3.33 bc	73.33 ± 3.33 b	93.33 ± 3.33
Chlorfenapyr	66.67 ± 3.33 bc	76.67 ± 3.33 ab	86.67 ± 3.33	53.33 ± 8.82 abc	66.67 ± 12.02 b	83.33 ± 3.33
Imidacloprid	93.33 ± 3.33 a	96.67 ± 3.33 a	100.00 ± 0.00	76.67 ± 8.82 a	90.00 ± 5.77 a	100.00 ± 0.00
Myclobutanil	76.67 ± 6.67 abc	83.33 ± 3.33 ab	93.33 ± 3.33	50.00 ± 5.77 abc	80.00 ± 5.77 ab	93.33 ± 3.33
Triadimefon	70.00 ± 5.77 bc	76.67 ± 3.33 ab	90.00 ± 5.77	53.33 ± 3.33 abc	76.67 ± 6.67 ab	86.67 ± 6.67
Azadirachtin	63.33 ± 3.33 bc	70.00 ± 0.00 b	93.33 ± 3.33	40.00 ± 5.77 c	63.33 ± 3.33 b	86.67 ± 8.82
Matrine	70.00 ± 5.77 bc	80.00 ± 5.77 ab	80.00 ± 5.77	50.00 ± 5.77 abc	76.67 ± 3.33 ab	83.33 ± 3.33
Osthole	93.33 ± 3.33 a	93.33 ± 3.33 ab	93.33 ± 3.33	56.67 ± 3.33 abc	80.00 ± 5.77 ab	93.33 ± 3.33
Pyrethrin	76.67 ± 3.33 abc	93.33 ± 3.33 ab	93.33 ± 3.33	66.67 ± 8.82 abc	80.00 ± 5.77 ab	83.33 ± 3.33
Rotenone	76.67 ± 3.33 abc	90.00 ± 5.77 ab	96.67 ± 3.33	40.00 ± 0.00 c	60.00 ± 5.77 b	86.67 ± 3.33
Chlorbenzuron	86.67 ± 3.33 ab	86.67 ± 3.33 ab	93.33 ± 3.33	40.00 ± 5.77 c	60.00 ± 5.77 b	93.33 ± 3.33
Diflubenzuron	76.67 ± 6.67 abc	90.00 ± 5.77 ab	96.67 ± 3.33	56.67 ± 8.82 abc	73.33 ± 3.33 b	86.67 ± 3.33
Flufenoxuron	73.33 ± 3.33 abc	83.33 ± 3.33 ab	90.00 ± 0.00	76.67 ± 3.33 ab	76.67 ± 3.33 ab	93.33 ± 3.33
Hexanumuron	90.00 ± 0.00 ab	90.00 ± 0.00 ab	93.33 ± 3.33	56.67 ± 3.33 abc	76.67 ± 8.82 ab	90.00 ± 5.77
EPN control	50.00 ± 5.77 c	86.67 ± 3.33 ab	93.33 ± 3.33	50.00 ± 5.77 abc	70.00 ± 5.77 b	90.00 ± 0.00

^1^ Means within a column followed by different letters are significantly different (*p* < 0.05, Tukey’s test).

**Table 3 insects-10-00161-t003:** The cost and benefit for the chive gnat control by EPN–imidacloprid combinations and chemicals in chive production in the year 2016–2017.

Treatment ^1^	EPN Rate (×10^9^ IJ·ha^−1^)	Imidacloprid Rate	Cost ^2^ (CNY·ha^−1^)	Chive Yield (kg·ha^−1^)	Chive Price (CNY·kg^−1^)	Income (CNY·ha^−1^)	Profit (CNY·ha^−1^)
Experiment conducted in Zhangqiu city							
LN2-imidacloprid	9	3 × 1/10 RC	5409	28,814	12	345,773	340,364
SN-imidacloprid	9	3 × 1/10 RC	9009	40,420	12	485,042	476,033
LN2 + SN + imidacloprid	4.5 + 4.5	3 × 1/10 RC	7209	39,920	12	479,039	471,830
chlorbenzuron	0	4 × 1 RC	2520	35,518	5.6	198,900	196,379
Control	0	0	0	26,813	12	321,761	321,761
Experiment conducted in Weifang city							
LN2-imidacloprid	6	2 × 1/10 RC	3606	44,022	3.8	167,284	163,678
SN-imidacloprid	6	2 × 1/10 RC	6006	45,623	3.8	173,367	167,361
LN2 + SN + imidacloprid	3 + 3	2 × 1/10 RC	4806	43,922	3.8	166,903	162,097
Phoxim	0	4 × 1 RC	1920	42,621	3.8	161,961	160,041
Control	0	0	0	24,412	3.8	92,766	92,766

^1^ LN2: *Heterorhabditis indica*; SN: *Steinernema feltiae*.^2^ Data were calculated based on the prices of EPN, insecticides, and chive in October 2017.

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
