# Peer review of "Integrated Management of Chive Gnats (*Bradysia odoriphaga* Yang & Zhang) in Chives Using Entomopathogenic Nematodes and Low-Toxicity Insecticides"

_insects, 2019, doi:10.3390/insects10060161_

Round 1

Reviewer 1 Report

Overall I have found this manuscript interesting. The combining of chemicals with EPN’s is an ever increasing subject area. However, I would like the authors to consider the following points:

Introduction:

The literature cited within the introduction is quite narrow. There are several studies that I am aware off where EPN’s have been successfully mixed with chemical insecticides for different pest species control e.g.  Rovesti and Desco (1990), Nematologia, 36: 237-245; Krishnayya and Grewal (2002), Biocontrol, Science and Technology, 12: 259-266; Cuthbertson et al. (2003), Nematology, 5: 713-720; Cuthbertson et al. (2007), Journal of Environmental Research and Development, 2: 1-5;  Cuthbertson et al., (2008), Insect Science, 15: 447-453. These references (and others) should be included in the introduction and/or the discussion section to compare the current study findings against in order to improve the breadth of the mansucript.

A few specific queries:

Line 40 - what does ‘occurs in a sheltered circumstance mean’?  Sentence needs rewording.

Line 42 – ‘heavy insecticide residues in the chives and fields’ - residues in the ‘fields’? – meaning what? Needs rewording.

Line 59 - what are ‘environmentally friendly’ insecticides? Is there such a thing?

Lines 74-75. More information required as to how the survival of the IJ’s was measured. Much work has been down in this area e.g. Fan and Hominick 1991, Review of Nematology 14: 381-387; Cuthbertson et al., 2003 Nematology.   Was a standard bioassay used?

Line 97 – how was the mortality of the IJ’s calculated – how were they counted?? Should a larval bioassay not have been done to test their infectivity?

Line 121: In the field experiments how robust are the EPN’s in the open field? Are they susceptible to UV etc.

Line 137 – what size of nozzle was on the irrigation system in order to prevent damage to the nematodes?

Line 284: on Figure 5 what does ‘CK’ stand for? Control?

In the discussion as with the introduction a wider comparison of the findings against what is already known is necessary. Increased efficacy of mixing S. feltiae with some chemicals (e.g. imidacloprid) is already known to occur.

The sentence structure and language of the manuscript requires some minor improvement.

Author Response

Point 1: Introduction: The literature cited within the introduction is quite narrow. There are several studies that I am aware off where EPN’s have been successfully mixed with chemical insecticides for different pest species control e.g. Rovesti and Desco (1990), Nematologia, 36: 237-245; Krishnayya and Grewal (2002), Biocontrol, Science and Technology, 12: 259-266; Cuthbertson et al. (2003), Nematology, 5: 713-720; Cuthbertson et al. (2007), Journal of Environmental Research and Development, 2: 1-5; Cuthbertson et al., (2008), Insect Science, 15: 447-453. These references (and others) should be included in the introduction and/or the discussion section to compare the current study findings against in order to improve the breadth of the manuscript.

Response 1: We have cited the references mentioned by the reviewer, in the Introduction and Discussion sections (Lines 57-63, 319-321, 330-335).

A few specific queries:

Point 2: Line 40 - what does ‘occurs in a sheltered circumstance mean’? Sentence needs rewording.

Response 2: Reworded (Line 40)

Point 3: Line 42 – ‘heavy insecticide residues in the chives and fields’ - residues in the ‘fields’? – meaning what? Needs rewording.

Response 3:residues in the fields” is changed to “residues in the soil” (Line 42)

Point 4: Line 59 - what are ‘environmentally friendly’ insecticides? Is there such a thing?

Response 4: Reworded in the text (Line 58)

Point 5: Lines 74-75. More information required as to how the survival of the IJ’s was measured. Much work has been down in this area e.g. Fan and Hominick 1991, Review of Nematology 14: 381-387; Cuthbertson et al., 2003 Nematology. Was a standard bioassay used?

Response 5: The method to measure the survival of the IJ is described and the relevant reference is cited (Lines 75-79). Reference for the bioassay method used is also cited in the text (Lines 104-106).

Point 6: Line 97 – how was the mortality of the IJ’s calculated – how were they counted?? Should a larval bioassay not have been done to test their infectivity?

Response 6: Method to count alive and dead IJ has been added in the text (Line 75-79). Infectivity of the IJ without insecticide treatment was evaluated in the bioassay against T. molitor and the result showed that the IJs were with high infectivity against T. molitor.

Point 7: Line 121: In the field experiments how robust are the EPN’s in the open field? Are they susceptible to UV etc.

Response 7: EPNs were applied in the field at cloudy days or after 4:00 pm to avoid strong UV radiation. This has been explained in the text (Line 138-139).

Point 8: Line 137 – what size of nozzle was on the irrigation system in order to prevent damage to the nematodes?

Response 8: The size of the nozzle was 6.5 mm in diameter, with which no shear damage will occur on the two EPN isolates. The size of the nozzle has been added in the text (Line 143-144, 153-154).

Point 9: Line 284: on Figure 5 what does ‘CK’ stand for? Control?

Response 9: CK stands for water control. The information is added in the caption of figure 4 and figure 5 (Line 264, 296).

Point 10: In the discussion as with the introduction a wider comparison of the findings against what is already known is necessary. Increased efficacy of mixing S. feltiae with some chemicals (e.g. imidacloprid) is already known to occur.

Response 10: The findings have been compared and the relevant references have been cited in the discussion (Line 319-321, 330-335).

Point 11: The sentence structure and language of the manuscript requires some minor improvement.

Response 11: Revised.

Reviewer 2 Report

This manuscript addresses compatibility of entomopathogenic nematodes with readily used insecticides to develop a sustainable management system to manage the chive gnat. If implemented the described protocols would benefit the growers and the environment. Granted there are several studies cited relating to this manuscript.

In applied entomology originality is rare. These sets of data add to the literature as described in 1. Its originality is in that a complete research project is reported instead of splitting into several piece-meal manuscripts.

The conclusions of this manuscript is consistent with the data.

Below is the minor comments on this manuscript:

Line 28, replace or reduce

Line 30, entries need to be italicized 

Line 61, give genus and species of white grubs

Lines 74-76, define technique to assess viability

Lines 78-83, reference rearing procedures

Line 93, size of Petri dish?

Line 101, give instar of wire-worm larvae

Line 117, how was synergism determined?

Line 195, replace caused with cause

Line 212, see 195

Line 224, no letters of significance on bars expressing data on 72 hr @ 25 degrees

Lines 291-298, what influences price of chives?

Author Response

#Reviewer 2

Below is the minor comments on this manuscript:

Point 1: Line 28, replace or reduce

Response 1: We have changed to “reduce” (Line 28).

Point 2: Line 30, entries need to be italicized

Response 2: Revised in the Keywords.

Point 3: Line 61, give genus and species of white grubs

Response 3: Added (Line 61-62).

Point 4: Lines 74-76, define technique to assess viability

Response 4: Defined (Line 75-79).

Point 5: Lines 78-83, reference rearing procedures

Response 5: References added (Line 80-87).

Point 6: Line 93, size of Petri dish?

Response 6: The size of the Petri dish used is 9 cm. Added in the text (Line 97).

Point 7: Line 101, give instar of wire-worm larvae

Response 7: 9th to 11th, size at 2-3 cm. Added in the text (Line 105).

Point 8: Line 117, how was synergism determined?

Response 8: Synergistic, additive, or antagonistic interactions between agents in the combination treatments were determined using a χ2 test with the control corrected data as described by Guo et al., 2017. This is explained in Section 2.6 (Line 171-173).

Point 9: Line 195, replace caused with cause

Response 9: Revised (Line 202).

Point 10: Line 212, see 195

Response 10: Revised (Line 219).

Point 11: Line 224, no letters of significance on bars expressing data on 72 hr @ 25 degrees

Response 11: No significant difference was found in the corrected mortality rates of the larvae at 72 h at 25 ºC among different treatments. So no letters were labeled on the bars.

Point 12: Lines 291-298, what influences price of chives?

Response 12: The field in Zhangqiu which we used to do the field experiments was qualified to plant pollution-free chives. So the chives harvest from this field could be sold at a higher price. The main factor that affect chive price is the quality of the chives. Price of chives with high quality could go up to 4 to 10 times of that of the common ones. The market demands, the chive supply and the brand value also affect the price of the chives.